# Reactivity of Rheumatoid Arthritis-Associated Citrulline-Dependent Antibodies to Epstein-Barr Virus Nuclear Antigen1-3

**DOI:** 10.3390/antib11010020

**Published:** 2022-03-11

**Authors:** Ilaria Fanelli, Paolo Rovero, Paul Robert Hansen, Jette Lautrup Frederiksen, Gunnar Houen, Nicole Hartwig Trier

**Affiliations:** 1Interdepartmental Laboratory of Peptide and Protein Chemistry and Biology, Department of NeuroFarBa, University of Florence, 50019 Sesto Fiorentino, Italy; ilaria.fanelli1@stud.unifi.it (I.F.); paolo.rovero@unifi.it (P.R.); 2Department of Drug Design and Pharmacology, University of Copenhagen, 2100 Copenhagen, Denmark; prh@sund.ku.dk; 3Department of Neurology, Rigshospitalet Glostrup, 2600 Glostrup, Denmark; jette.lautrup.battistini@regionh.dk; 4Department of Biochemistry and Molecular Biology, University of Southern Denmark, 5230 Odense M, Denmark

**Keywords:** anti-citrullinated protein antibodies, citrullinated peptides, Epstein-Barr nuclear antigen, Epstein-Barr virus, multiple sclerosis, myelin basic protein, rheumatoid arthritis, rheumatoid factor

## Abstract

Rheumatoid arthritis (RA) is a chronic disease which causes joint inflammation and, ultimately, erosion of the underlying bone. Diagnosis of RA is based on the presence of biomarkers, such as anti-citrullinated protein antibodies (ACPA) and rheumatoid factors, along with clinical symptoms. Much evidence points to a link between the Epstein-Barr virus and RA. In this study, we analyzed ACPA reactivity to citrullinated peptides originating from Epstein-Barr nuclear antigens (EBNA1, EBNA2, and EBNA3) in order to elaborate the diagnostic potential of citrullinated EBNA peptides. Moreover, ACPA cross-reactivity to citrullinated peptides from myelin basic protein (MBP) was analyzed, as citrullinated MBP recently was described to be associated with multiple sclerosis, and some degree of sequence homology between MBP and citrullinated EBNA exists. A peptide from EBNA2, (EBNA2-A, GQGRGRWRG-Cit-GSKGRGRMH) reacted with approximately 70% of all RA sera, whereas only limited reactivity was detected to EBNA1 and EBNA3 peptides. Moreover, screening of ACPA reactivity to hybrid peptides of EBNA3-A (EPDSRDQQS-Cit-GQRRGDENRG) and EBNA2-A and peptides containing citrulline close to the N-terminal confirmed that ACPA sera contain different populations of ACPAs. No notable ACPA reactivity to MBP peptides was found, confirming that ACPAs are specific for RA, and that other factors than the presence of a central Cit-Gly motif are crucial for antibody binding. Collectively, these findings illustrate that citrullinated EBNA2 is an optimal candidate for ACPA detection, supporting current evidence that EBV is linked to RA onset.

## 1. Introduction

Rheumatoid arthritis (RA) is a chronic systemic disease, which is characterized by inflammation in the synovial tissue of joints [1,2]. In industrialized countries, 0.5–1% of adults suffer from RA, with a significant reduction of life quality [2]. RA is two to three times more common in women than men, and disease incidence peaks between 40 to 60 years [3]. In the earlier stages of RA, symptomatology is characterized by morning stiffness, tender and swollen joints, flu-like feeling, and fatigue, while cartilage destruction, joint misalignment, and bone erosion are common in the later stages [1,4]. Additionally, RA may have systemic manifestations, such as lung disease, pleural effusions, vasculitis, keratoconjunctivitis, atherosclerosis, and lymphomas [2,5].

RA is diagnosed according to the EULAR/ACR classification criteria revised in 2010, which, beyond clinical disease manifestations, comprise serological biomarkers, such as rheumatoid factors (RF) and anti-citrullinated protein antibodies (ACPA) [1,3,6]. While RFs IgA and IgM isotypes are directed to the Fc region of IgG, ACPAs are mainly of the IgG isotype, and are specific for citrullinated targets [7,8]. RFs are found approximately in 60–80% of RA cases, and occasionally in healthy individuals and in individuals suffering from systemic lupus erythematosus (SLE), Sjogren’s syndrome (SjS), or affected by infections [9,10,11]. In contrast, ACPAs are typically detected in 70–80% of RA patient sera, and are more disease-specific for RA compared to RFs, although ACPA seropositivity of 9% has been reported in other rheumatic diseases than RA [12,13,14,15,16]. Moreover, ACPAs have been detected up to 14 years before the clinical symptoms of RA [17,18].

ACPAs are specific for citrulline, a non-genetically encoded amino acid, which is the result of a post-translational modification catalyzed by peptidyl arginine deiminase (PAD) enzymes. PADs are calcium-dependent metalloenzymes, which are inactive at resting levels of intracellular calcium, and are activated with high levels of calcium [19]. The conversion of the guanidino group of Arg to the neutral ureido group of citrulline leads to loss of a positive charge and change of the isoelectric potential, which may result in a structural unfolding [20,21]. Under natural conditions, citrullination of proteins occurs in cells undergoing apoptosis, where citrullinated proteins are marked for degradation and are cleared from the body by engulfment of the apoptotic bodies by phagocytes. In RA, the citrullinated proteins are not properly cleared, and consequently exposed to the immune system, resulting in generation of ACPAs [20].

ACPAs are commonly detected in commercially available immunoassays employing citrullinated peptides. The first generation of ACPA assays was based on a synthetic linear citrullinated peptide derived from human filaggrin [8]. In order to improve the sensitivity of the assay, a cyclic citrullinated peptide (CCP) was introduced, as it yielded higher sensitivity and specificity relative to the linear pro-filaggrin peptide [8,22]. Screening of peptide libraries led to the selection of other further antigens, which resulted in the second and third generation of assays, CCP2 and CCP3, respectively [23,24,25,26].

In the commercial ACPA detection assays, different citrullinated peptides are employed, which is in accordance with the cross-reactive nature of ACPAs. ACPA often recognize several citrullinated targets, preferably containing a Cit-Gly motif [8,12,13,17,27,28,29,30,31]. Examples of ACPA targets that have been reported are collagen, fibrinogen, α-enolase, vimentin, pro-filaggrin, Epstein-Barr nuclear antigen (EBNA)1, and EBNA2 [12,13,27,29,32]. With the exception of the Cit-Gly motif, studies indicate that substitutions in positions close to the motif do not influence antibody reactivity. These findings confirm the crucial role of the Cit-Gly motif for a stable antibody-antigen interaction, even though other amino acids besides Gly are occasionally tolerated [13,28,30,31,33].

It has been debated whether ACPAs are associated with the pathogenesis of RA, even though the etiology of RA still is unclear [2]. Currently, it is believed that a combination of genetic and environmental factors contributes to RA onset [34]. The genetic factors with the highest association for developing RA are the human leukocyte antigen (HLA)-DRB1 genes in the locus of class II MHC and non-MHC genes [35,36,37,38]. The most closely associated with RA are HLA-DRB1*01 and HLA-DRB1*04 alleles, which all share a sequence of five amino acids, known as the shared epitope (SE), and are responsible for antigen presentation to T-cells [39].

In relation to environmental triggering factors, smoking appears to contribute to RA onset. Smoking may be involved in the activation of PAD enzymes, which results in the citrullination of proteins, leading to the presentation of citrullinated antigens, and ultimately the generation of ACPAs [40,41]. Moreover, Epstein-Barr virus (EBV) infection has been proposed to be involved in RA onset [42,43,44]. EBV is a common virus, which has been proposed to be associated with a number of rheumatic autoimmune diseases, e.g., SjS and SLE, but also other autoimmune diseases, such as multiple sclerosis (MS) [42,45]. EBV targets various cells and regions, depending on the specific disease [42,45]. The possible implication of EBV in RA was originally proposed by Alspaugh and Tan, who showed elevated antibody titers to EBNA1 in sera from RA patients [46]. Later, it was described that RA sera reacted with EBV viral capsid antigen (VCA), early antigen (EA) and EBNA2 [47,48,49]. Additionally, it has been found that RA patients have elevated EBV DNA levels in peripheral blood mononuclear cells, with a 10-fold increase in circulating EBV-infected B cells in saliva and the synovial fluid compared to healthy individuals [47,49,50]. These findings are in accordance with recent studies, describing that EBV gp42, a glycoprotein fundamental for EBV binding and entry into B cells, is an optimal ligand for HLA-DRB1 SE alleles [51], and that EBNA2 is inclined to bind to RA-associated genetic loci [52,53].

In addition to genetic and virological associations, serological data strengthen the hypothesis of a link between EBV and RA [53]. Studies illustrate that some citrullinated peptides originating from EBNA1 and EBNA2 are good candidates for ACPAs detection. In particular, an EBNA2 peptide obtains a sensitivity comparable to current diagnostic assays, and shows strong sequence similarity to the pro-filaggrin peptide originally used for ACPA identification and RA diagnosis [8,12,27].

In the current study, we analyzed the reactivity of RA sera to citrullinated EBV peptides from various EBNA proteins, to elaborate the diagnostic potential of citrullinated EBNA peptides, and to investigate the potential link between EBV and RA further.

## 2. Materials and Methods

### 2.1. Reagents

Alkaline phosphatase (AP)-conjugated goat-anti-human IgG, streptavidin, AP substrate tablets (*para*-nitrophenylphosphate (*p*NPP)), Tris, Tween, sodium chloride, sodium carbonate, ethanolamine, MgCl_2_, D-Biotin, hexafluorophosphate azabenzotriazole tetramethyl uranium (HATU), milliQ water, trifluoroacetic acid (TFA), triisopropylsilane (TIS), and *N,N*-diisopropylethylamine (DIPEA) were from Sigma Aldrich (St. Louis, MO, USA). Synthetic EBNA1 and EBNA2 peptides were purchased from Schäfer-N (Lyngby, Denmark) (Table 1), and were generated on TentaGel resin using standard Fmoc-based solid-phase peptide synthesis (SPPS) [54]. The peptides were synthesized as peptide acids. EBNA3 peptides were synthesized by microwave-assisted SPPS [55,56]. Wang resin, Oxyma Pure, and DIC was from Iris Biotech AG (Marktredwitz, Germany). Fmoc-L-amino acids were from Novabiochem (Darmstadt, Germany). Acetonitril was from Carlo Erba (Milano, Italia).

### 2.2. Peptides

Citrullinated peptides from EBNA1, EBNA2, EBNA3, and MBP were tested for reactivity (Table 1). Peptides from EBNA1 and EBNA2 were selected based on previous studies, where the three peptides with the highest sensitivity from each protein were selected for further testing [8,27]. EBNA3 and MBP peptides were selected based on the presence of naturally occurring Arg-Gly motifs, where arginine was substituted with citrulline. All of the peptides screened were 20 amino acids long, and contained a Cit-Gly-Xxx or Gly-Cit-Gly motif.

### 2.3. Generation of Biotinylated Synthetic EBNA3 Peptides

Citrullinated peptides originating from EBNA3 were synthesized under standard protocols by automated microwave-assisted SPPS using the “Liberty Blue” by CEM corporation (Matthews, NC, USA) [56]. Pre-loaded Wang resins were used for synthesis. Oxyma Pure and DIC were utilized for activation, and a single coupling for each amino acid was performed. After peptide synthesis, each peptide was conjugated with biotin in the N-terminal. To this purpose, a portion of the resin was swollen in DMF and treated with D-Biotin (five equivalents), DIPEA (seven equivalents), and HATU (five equivalents) for 40 min at room temperature (RT) under magnetic stirring.

Cleavage of the peptides, along with the removal of side-chain protecting groups, was conducted using 95% TFA, 2.5% MilliQ water and 2.5% TIS. The resins were treated with the cleavage mixture under magnetic stirring for 90 min, whereafter the solution containing the cleaved peptide was filtered and treated with isopropyl ether at 4 °C to precipitate the crude peptide. The filtered solution was centrifuged at 3500 rpm (2 × 5 min), whereafter the supernatant was discarded. The pellet was vacuum-dried and dissolved in 50:50 ACN/MilliQ water. Following peptide lyophilization, the peptides were characterized using reverse phase ultra-high-performance liquid chromatography and electro-spray ionization mass spectrometry (Ultimate 3000—Thermo Fisher Scientific, Waltman, MA, USA).

### 2.4. Patient Material

RA serum samples (*n* = 28) and healthy donor (*n* = 28) serum samples (referred to as healthy controls (HC)) were from the Biobank at Statens Serum Institut (Copenhagen, Denmark). The samples were tested anonymously, therefore not requiring ethical consent. MS serum samples (*n* = 7) were from the Multiple Sclerosis Clinic, Department of Neurology, Rigshospitalet Glostrup (Glostrup, Denmark). The samples were tested anonymously, therefore did not require ethical consent.

### 2.5. Detection of Antibodies by Enzyme-Linked Immunosorbent Assay and Streptavidin-Capture Enzyme-Linked Immunosorbent Assay

Microtiter plates were coated with free peptide (1 µg/mL) in carbonate buffer (0.05 M sodium carbonate, pH 9.6), and incubated overnight at RT on a shaking table (ST). All blocking (30 min), washing (3 × 1 min), and incubation (1 h) steps were conducted using TTN (20 mM Tris, 0.01% Tween 20, 0.3 M NaCl, pH 7.5) buffer. Wells were rinsed and blocked, whereafter sera (1:200 dilution) were added to each well, and incubated for at RT on a ST. Next, wells were washed, and AP-conjugated goat-anti-human IgG (1 µg/mL) was added to each well and incubated at RT on a ST. Finally, AP substrate buffer (1 M ethonolamine, 0.5 mM MgCl_2_, pH 9.8) with *p*NPP (1 mg/mL) was added to each well, and AP activity was determined by measuring the absorbance at 405 nm with background subtraction at 650 nm (Versa Max plate reader, Molecular Devices, San Jose, CA, USA).

Alternatively, microtiter plates were precoated with streptavidin (1 µg/mL) in carbonate buffer, and incubated overnight at 4 °C. Biotinylated peptides (diluted 1 µg/mL in carbonate buffer) were added to each well, and incubated for 2 h at RT on a ST. The following steps in the experiment were carried out as described above.

Absorbances of all the results were normalized to a positive RA control pool (*n* = 20), in order to ensure that all results were directly comparable. All samples in the RA positive pool were selected based on a high ACPA titer. A peptide-specific cut-off was introduced based on preliminary screenings, tolerating a non-specific reactivity of 5% and an intra assay variation of maximum 15%. Readings above the cut-off were regarded as positive, whereas samples below the cut-off were regarded as being negative. Inter assay variations below 15% were acceptable. All samples were tested in duplicate.

### 2.6. Data Interpretation and Statistics

Statistical analyzes and plots were generated using Graph Pad Prism 5.0 software. Samples were normalized relative to a positive control, allowing a specificity of 95%. Data are presented as mean ± standard deviation. Using this software, statistical significance was assessed by nonparametric unpaired two-tailed Mann–Whitney U-test. Significant differences are indicated by *: *p* < 0.05, **: *p* < 0.01, ***: *p* < 0.001.

## 3. Results

### 3.1. Reactivity of Rheumatoid Arthritis Sera and Healthy Control Sera to Citrullinated Epstein-Barr Nuclear Antigen Peptides

To analyze ACPA reactivity to citrullinated EBNA peptides, RA and HC sera were screened for reactivity to citrullinated peptides originating from EBNA1, EBNA2, and EBNA3. Antibody reactivity was tested to 11 biotinylated EBNA peptides, 3 EBNA1 peptides, 3 EBNA2 peptides, and 5 EBNA3 peptides by ELISA.

Figure 1a illustrates the reactivity of RA sera to citrullinated EBNA1 peptides. In total, 39% of the RA sera reacted to EBNA1-A, and RA reactivity to EBNA1-A was significantly elevated when compared to HCs (*p* = 0.0029). No significant difference in RA reactivity was found to EBNA1-B and EBNA1-C compared to HC samples (*p* = 0.6310 for EBNA1-B and *p* = 0.2044 for EBNA1-C), as only 11 and 18% of RA sera reacted to EBNA1-C and EBNA1-B, respectively. In total, 50% of the RA sera reacted to one of the EBNA1 peptides, whereas 15% reacted to two peptides, and only one RA sample reacted to all three EBNA1 peptides. A total of 28% of RA samples tested did not react with any of the EBNA1 peptides tested.

In contrast, 75% of the RA sera reacted to at least one of the citrullinated EBNA2 peptides, whereas 31% reacted to two peptides (primarily EBNA2-A and EBNA2-C) (Figure 1b), and 25% of sera tested did not react with any of the EBNA2 peptides. The EBNA2-A peptide obtained a sensitivity of 68%, whereas 14% and 21% of the RA sera reacted to EBNA2-B and EBNA2-C, respectively. Only statistically significant antibody reactivity was determined for EBNA2-A compared to the HCs (*p* < 0.0001 for EBNA2-A, *p* = 0.3844 for EBNA2-B and *p* = 0.0977 for EBNA2-C).

Finally, RA reactivity to citrullinated EBNA3 peptides was analyzed. Limited antibody reactivity was detected, as only 4–7% of the RA sera reacted to the EBNA3-1, 2, 4 and 5 (Figure 1c). A total of 19% of the RA sera reacted to at least one of the five EBNA3 peptides.

Moreover, none of the HC samples showed specific reactivity to the 11 citrullinated EBNA peptides tested (Figure 1a–c).

When comparing ACPA reactivity to the EBNA peptides, 86% of the RA samples reacted to at least one of the citrullinated peptides, whereas 14% of the samples did not. When comparing between the most reactive EBNA peptides (EBNA1-A, EBNA2-A, EBNA3-E), EBNA2-A obtained the highest sensitivity of all peptides tested (68%) (Figure 1d). ACPA reactivity to EBNA2-A was significantly elevated compared to EBNA1-A (*p* < 0.0001), yielding a sensitivity of 39%, and to EBNA3-E (*p* = 0.0067). Moreover, 29% of the RA sera reacted with EBNA1-A and EBNA2-A; however, no correlation in antibody reactivity could be determined between the two peptides (r = 0.1030) (results not shown).

Collectively, these results illustrate that RA patient sera have specific antibody reactivities to citrullinated EBNA peptides. Moreover, citrullinated EBNA2-A is significantly recognized by RA sera when compared to the remaining EBNA peptides examined, making the EBNA2-A an important ACPA substrate, compared to the remaining EBNA peptides tested.

It has previously been proposed that amino acids C-terminal to citrulline are essential for ACPA reactivity [13,28,31]. To analyze this further, overcrossing peptides of EBNA3-A and EBNA2-A were tested for reactivity. For this, two peptides were synthesized, one containing the N-terminal of EBNA3-A and the C-terminal of EBNA2-A (EPDSRDQQS-Cit-GRSKGRGRMH), and one with the N-terminal of EBNA2-A and the C-terminal of EBNA3-A (GQGRGRWRG-Cit-GQRRGDENRG). In addition to this, an EBNA-2 peptide (NCit-EBNA2), containing Cit close to the N-terminal, was tested exposing the C-terminal end (G-Cit-GSKGRGRMHKLPEPRRPGPD). RA (*n* = 21), HC (*n* = 15) and MS (*n* = 7) sera reactivity to the selected peptides was determined in a direct ELISA using peptides coating directly to the polystyrene surface of microtiter plates.

As presented, RA sera reacted to all of the peptides tested (Figure 2a) compared to HC and MS samples (Figure 2b,c), where only sporadic reactivity was found, confirming that ACPA reactivity is specific for RA.

No significant difference in antibody reactivity was found to the overcrossing peptides (*p* = 0.0575 for EBNA3-EBNA2, *p* = 0.7504 for EBNA2-EBNA3) compared to EBNA2-A, although a trend indicated a reduced reactivity to EBNA3-EBNA2. In fact, the EBNA2-EBNA3 peptide reacted with 76% of the RA samples tested, whereas 67%, 47%, and 38% of RA sera reacted to EBNA2-A, EBNA3-EBNA2, and EBNA3-A, respectively, indicating that this overcrossing peptide is a slightly better substrate than EBNA2-A in this assay. When testing for possible correlations between the antibody reactivities, a moderate negative correlation was found between EBNA2-A and EBNA2-EBNA3 (r = −0.4722) (*p* = 0.0307) (Figure 2d) (results not shown).

In contrast to EBNA2-A, antibody reactivity to EBNA2-EBNA3 was significantly elevated when compared to EBNA3-A (*p* = 0.0091), whereas no significant difference in antibody reactivity was found between EBNA3-A and EBNA3-EBNA2 (*p* = 0.7066).

As presented, antibody reactivity to NCit-EBNA2-A was significantly reduced compared to EBNA2-A (*p* = 0.0022), indicating that other factors than the presence of the C-terminal of EBNA2-A in combination with the crucial Cit-Gly motif are essential for antibody reactivity.

Collectively, these findings confirm that other factors than just the mere presence of a Cit-Gly motif in combination with specific amino acids C-terminal to the motif are essential for antibody reactivity.

### 3.2. Serologic Correlations between Rheumatoid Factors and EBNA-Specific Proteins and Peptides

To determine a potential correlation between serologic RA biomarkers and EBV antibodies, RA and HC samples were tested for reactivity to full-length EBNA1 and EBNA2 in ELISA.

As presented in Figure 3, all samples tested positive for IgG reactivity to full-length EBNA1 and EBNA2. No significant difference in EBNA IgG was detected when comparing RA reactivities to HCs (*p* = 0.8159 for EBNA1, *p* = 0.9536 for EBNA2). Nevertheless, RA EBNA1 IgG level was significantly elevated compared to EBNA2 IgG for RA and HCs (*p* < 0.0001).

Next, the serologic levels of the biomarkers RF IgA and IgM were analyzed by ELISA (Figure 4). In this, 96% and 92% of the RA samples were positive for RF IgA and RF IgM, respectively; moreover, two HCs tested positive for RF IgA and IgM.

A mean of 3237 IU/mL was found for IgM RFs for RA sera, which was significantly elevated compared to HC sera with an average of 64 IU/mL (*p* < 0.0001). Moreover, a moderately positive correlation was determined between IgM and IgA RFs titers (r = 0.5979) (*p* = 0.0016).

No correlation was determined between EBNA1 IgG and IgA RF (r = 0.006223), EBNA1 IgG and IgM RF (r = −0.06233), EBNA2 IgG and IgA RF (r = −0.02009), and EBNA2 IgG and IgM RF (r = −0.1015) (results not shown).

In addition to full-length proteins, a potential connection between RFs and EBNA2-A was examined, which was the peptide with the highest sensitivity among the peptides screened (Table 2).

As presented in Table 2, when comparing “single positives”, 76% of the RA samples were positive to either EBNA2-A, RF IgM or RF IgA, and all RA samples were positive for at least one of the three antibodies. When comparing double positive samples, at least 72% of RA samples reacted to EBNA2-A in combination with either RF IgM or RF IgA, whereas 88% of samples tested positive for both RFs. Finally, 68% of RA samples were positive for reactivity to EBNA2 and RF IgM and IgA, although no correlation was determined between EBNA2-A and IgA RF (r = −0.06240) and EBNA2-A and IgM RF (r = −0.06283) (results not shown). Ultimately, these studies confirm the presence of ACPAs in RA sera that are specific for citrullinated peptides originating from EBNA2, although no correlation between ACPA-specific reactivity to EBNA2-A and RF titers was determined.

### 3.3. Anti-Citrullinated Protein Antibody Reactivities to Citrullinated MBP Peptides

Recently, it was reported that elevated citrullination levels are found in MS [57,58]. To analyze this further, citrullinated Arg-Gly peptides originating from MBP, an auto-antigen associated to MS, were synthesized and analyzed for RA, HC and MS reactivity in ELISA. In total, four citrullinated MBP peptides were tested for reactivity. In particular, peptide 4 was of interest, since it contains sequence similarity to the EBNA2-A peptide with the highest sensitivity for ACPA.

Figure 5 illustrates the reactivity of RA, MS, and HCs to MBP peptides and EBNA2-A. As seen, a few RA sera reacted to the MBP peptides, whereas no reactivity was observed for MS and HC sera, confirming that reactivity to citrullinated peptides is specific for RA. Collectively, although a central Cit-Gly motif was present in all of the MPB peptides, none of the citrullinated peptides were good substrates for ACPA detection.

## 4. Discussion

RA is a chronic inflammatory disease, where the serologic biomarkers ACPA and RF are central for the diagnosis and monitoring of RA. A total of 40 to 60% of early RA cases are positive for RF, indicating that RF appears before the clinical symptoms, although the specificity of RF is not impressive [9]. In contrast, ACPA levels are a specific indicator for the early prediction, identification, and diagnosis of RA; moreover, ACPA levels are significantly related to bone erosion [59,60]. Based on this knowledge, we examined ACPA binding to various citrullinated peptides and RF levels in serum of RA and HCs, and to elaborate on the diagnostic potential of citrullinated EBNA peptides. Citrullinated EBNA peptides were selected for further analysis based on several findings, indicating a link between EBV and RA.

A total of 11 peptides originating from different EBNA proteins were tested for reactivity with RA, HC, and MS samples. Whereas three EBNA1 and three EBNA2 peptides had been previously tested [17,27], antibody reactivity to the five EBNA3 peptides were analyzed for the first time. It has been reported that the EBNA1 peptides obtained a sensitivity of 10–47%, and that EBNA1-B was the most sensitive candidate for ACPA detection, compared to EBNA1-A [17], while EBNA1-A obtained higher sensitivities than EBNA1-B in the current study (Figure 1). Regarding the EBNA2-A peptide, a sensitivity of approximately 67% was found, although it has been reported to achieve a sensitivity of 93%, using a larger RA cohort [27]. Collectively, these findings confirmed that the EBNA2-A peptide is an important substrate for ACPA detection, which previously has been reported to be able to compete with commercially available ACPA detection kits [27]. Interestingly, the biotinylated citrullinated EBNA3 peptides were classified as poor ACPA substrates. These findings are in accordance with the literature, describing that the presence of a Cit-Gly motif not is a guarantee for ACPA reactivity [20,28], other factors contribute to reactivity, e.g., peptide structure, as originally reported by Schellekens and colleagues [22].

In fact, screening of antibody reactivity to NCit-EBNA2, overcrossing EBNA peptides and MPB peptides, confirmed that other factors than the presence of Cit-Gly are essential for ACPA reactivity. Although NCit-EBNA2 and EBNA1-A contained a Cit-Gly motif in the same position, ACPA reactivity to NCit-EBNA2 was reduced when compared to EBNA1-A and EBNA2-A (Figure 1 and Figure 2). A possible explanation could very well be that a specific secondary structure is required for optimal presentation of the Cit-Gly motif, as originally described by Schellekens [8]. These findings are in accordance with results describing ACPA reactivity to overcrossing EBNA peptides (Figure 2) and MBP peptides (Figure 5). Preliminary studies have indicated that the C-terminal of citrullinated peptides often is important, especially if a charged amino acid is present, e.g., Arg or Lys. Based on this, one would expect the EBNA3-EBNA2 peptide, to have the highest sensitivity of the overcrossing peptides. However, screening of EBNA overcrossing peptides showed that the EBNA2-EBNA3 peptide obtained a high sensitivity compared to the EBNA3-EBNA2 peptide (76% vs. 47%). These findings are in accordance with results obtained from screening of the NCit-EBNA2 peptide.

Collectively, results obtained support that other factors than the Cit-Gly motif, e.g., secondary structures are essential for ACPA reactivity. This remains to be elaborated; for instance, it would be interesting to evaluate the folded structure of the peptides by performing circular dichroism analyzes.

Another explanation to the difference in reactivity is the presence of overlapping ACPA populations. ACPA has previously been described to contain a group of overlapping and non-overlapping reactivities, which most likely is ascribed to backbone and side-chain dependent antibody reactivity, respectively [20,28,61]. Based on this, one would expect that the ACPAs recognizing EBNA2-A, which do not recognize EBNA2-EBNA3, to be side-chain dependent, whereas the ones reacting to this peptide are backbone-dependent. This remains to be elaborated. Interestingly, however, a moderate negative correlation (r = −0.4722) was found between EBNA2-A and EBNA3-EBNA2 reactivity, which partially confirms the above-mentioned hypothesis.

Comparing the ACPA reactivity pattern to the EBV cycle transcription program, our results regarding EBNA3 peptides are surprising, because while EBNA1 is expressed throughout the lifecycle of the virus, EBNA2 and EBNA3 proteins are all expressed during the latency III stage, where the growth transcription program occurs [48]. For this purpose, as the presence of antibodies directed to EBNA2 had already been reported in the literature, thus showing the activation of the EBV growth program, we would have expected ACPA reactivity to EBNA3 peptides as well. Paying attention to the EBNA3 peptide sequences, it can be noticed that most of them have positively charged amino acids on the citrulline C-terminal, like EBNA2-A, which has been found to favor the antigen-antibody interaction and thus to increase the ACPA reactivity [30]. However, it appears that even if, in theory, these peptides have common characteristics to the well-recognized EBNA2-A peptide, these sequence homologies are not enough to guarantee a stable antigen-antibody interaction, confirming therefore that other factors are influencing ACPA reactivity.

As the peptides from EBNA1 and EBNA2 performed well in our assays, the RA and HC cohorts were tested for reactivity to full-length EBNA1 and EBNA2 proteins, and elevated absorbances were found for EBNA1 protein compared to EBNA2 (Figure 3). This result conforms to the literature, where it has been reported that EBNA1 antibodies persist at a stable titer for lifetime, while EBNA2 antibodies are the first to be detected, but then decline within weeks [43]. Nevertheless, no significant difference in reactivity was revealed between the RA and HC sera. This may be due to the worldwide prevalence of EBV infection, although previous studies reported higher levels of antibodies against EBV-encoded proteins in patients with RA compared to HCs [43,48,62]. Collectively, given the hypothesis that EBV may be involved in the onset of RA, it is surprising that a significant difference between the RA and the HCs cohorts was found when testing the EBNA peptides (EBNA1-A and EBNA2-A), but no difference between RA and HC sera was detected while testing EBNA full-length proteins. This may be ascribed to the small size of the cohort tested, and remains to be analyzed using a larger cohort, as studies in the literature describe that RA sera have higher EBNA1 titers compared to HCs [62].

RF titers of IgM and IgA isotypes were determined and sensitivities of 96% and 92% in the RA cohort were revealed, respectively (Figure 4). RF titers were compared to EBNA1 and EBNA2 full-length proteins and no correlation was evidenced as well, as no correlation was obtained between RFs titers and EBNA2. Nevertheless, approximately 75% of the RA sera were positive for either RFs or ACPAs specific for citrullinated EBNA2-A.

Finally, screening of MBP peptides with MS, RA and HC sera illustrated that ACPA reactivity is specific for RA as no antibody reactivity to citrullinated MBP was found in the MS sera tested, although it recently was described that MBP was citrullinated in MS disease [57,58]. 

In general, autoimmune rheumatic diseases proposed to be associated with EBV do not express ACPA, or at least only in very low levels [16]. Whether this occasional ACPA seropositivity indicates a false positive result or an overlap RA syndrome remains to be determined.

Previously, ACPA seropositivity of SLE and SjS samples to the selected EBNA1 peptides, tested in this study, were determined [12,17]. Specificities ranged from 93–100%, confirming that ACPA is specific for RA and only sporadically found in other non-rheumatic autoimmune diseases, which is in accordance with findings described in this study [12,17]. In addition to this, the specificity of EBNA2-A was determined in a recent study as well, by screening samples from individuals with SLE, SjS and HCs [27]. The EBNA2 peptide obtained a high specificity and was able to compete with commercial CCP assays, similar to the EBNA1 peptides [12,17,27]. In both studies, the EBNA1 peptides and the EBNA2 peptide obtained sensitivities and specificities similar to the CCPlus and CCP3.1 assay.

Speaking of diagnostic tools, several assays for ACPA and RF determination exist [10,63]. As a natural consequence, they vary in assay sensitivity and specificity. Where the commercial ACPA assays apply several citrullinated sequences, we only determined ACPA reactivity to the individual peptides in this study, which may improve the assay specificity. However, we only detected IgG ACPA, whereas the CCP3.1 detects both IgA and IgG ACPAs. Compared to RF, ACPA detection appears more sensitive and specific, although ACPA occasionally has been reported to be moderately sensitive dependent on cohort and assay [64]. As a consequence, both serologic biomarkers are part of the diagnostic classification criteria. RF have a high sensitivity for RA, but for other rheumatic and non-rheumatic diseases as well [63]. For example, RFs have been reported in high frequencies in SjS (75–95%), biliary cirrhosis (45–70%), HCV infection (40–76%), and mixed connective tissue diseases (50–60%). In contrast to ACPA, IgA and IgM RFs are detected, whereas mainly ACPA IgG is detected in CCP assays. In a clinical practice, it is still recommended to measure both biomarkers, as their combination improved diagnostic accuracy, especially in the case of early RA.

Another thing separating RFs and ACPAs is their association with disease course. Individuals with ACPA-positive RA often experience a progressive disease course with cartilage destruction and bone erosion compared to ACPA-negative RA, which often is associated with a milder disease outcome, and may indicate that ACPAs in theory are pathogenic, although the pathogenesis of ACPA still remains to be determined [2,65,66]. Some studies indicate that ACPAs are pathogenic, e.g., it has been demonstrated that the transfer of antibodies to citrullinated collagen or fibrinogen aggravate inflammatory arthritis in animal models [67]. Moreover, it has been proven that ACPAs can activate the complement system and recognize the FcR-positive cells [68]. In contrast, ACPA has been described to be present a decade before actual symptoms, which does not support a possible role in disease pathogenesis. For this reason, it has been suggested that in the inflamed synovium ACPAs are correlated to citrullinated antigens forming immune complexes and thus promoting the phlogoses’ progression. These findings are in line with the clinical manifestation of a poorer disease outcome in individuals with ACPA-positive RA compared to ACPA-negative patients [69]. Moreover, the occurrence of ACPA-positive RA is related to genetic risk factors that predispose for RA, for instance, protein phosphatase non-receptor type-22 (PTPN22) and MHC class II alleles, and exogenous factors, such as smoking and bacterial infections, which may promote deamination of arginine, through activation of PAD, resulting in the generation of citrullinated antigens and thus in the induction of ACPAs production [70,71]. Thereby, ACPAs are good predictors for poor RA prognoses [2].

Collectively, these findings indicate that citrullinated EBNA2-A in combination with RFs may provide a good tool for serologic diagnostics of RA, and that EBV to some extent may contribute to the onset of RA. Moreover, current findings indicate that secondary factors and the optimal location of Cit may vary between peptides of different origin. Although current findings confirm that ACPAs are specific for RA and good diagnostic markers, the actual role of ACPA still remains to be determined in detail.

## Figures and Tables

**Figure 1 antibodies-11-00020-f001:**
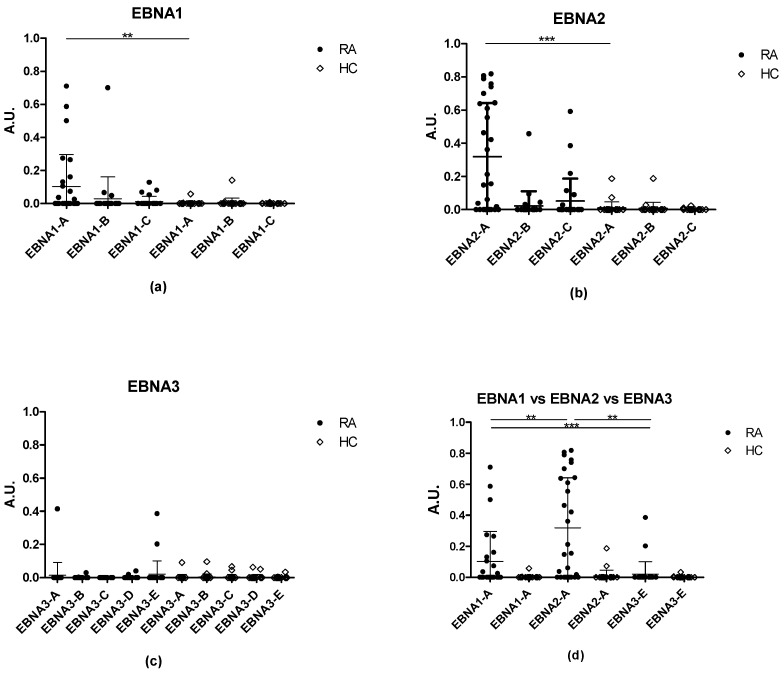
Reactivity of rheumatoid arthritis (RA) samples (*n* = 28) and healthy (HC) samples (*n* = 28) to biotinylated citrullinated EBNA1, EBNA2, and EBNA3 peptides: (**a**) Reactivity of RA and HC samples to EBNA1 peptides; (**b**) Reactivity of RA and HC samples to EBNA2 peptides; (**c**) Reactivity of RA and HC samples to EBNA3 peptides; (**d**) Reactivity of RA and HC samples to the most reactive EBNA peptides. ** = *p* < 0.05, *** = *p* < 0.001.

**Figure 2 antibodies-11-00020-f002:**
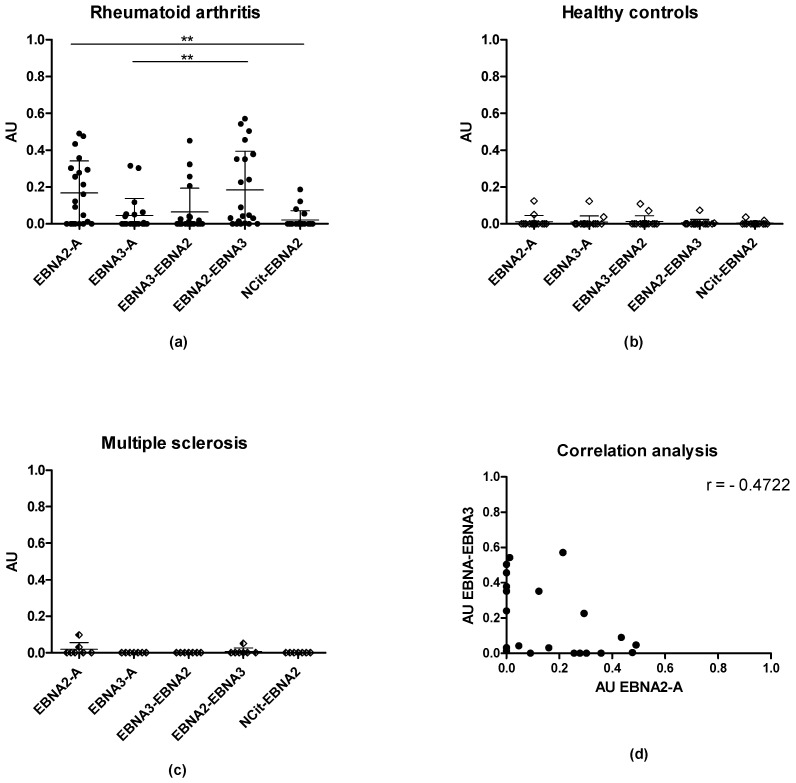
Reactivity of rheumatoid arthritis (RA) sera, multiple sclerosis (MS) sera and healthy control (HC) sera to overcrossing and N-terminal citrullinated peptides analyzed by enzyme-linked immunosorbent assay: (**a**) reactivity of RA sera to citrullinated peptides; (**b**) reactivity of HC sera to citrullinated peptides; (**c**) reactivity of MS sera to overcrossing peptides; (**d**) correlation analysis of RA antibody reactivity between EBNA2-A and EBNA2-EBNA3. ** = *p* < 0.01.

**Figure 3 antibodies-11-00020-f003:**
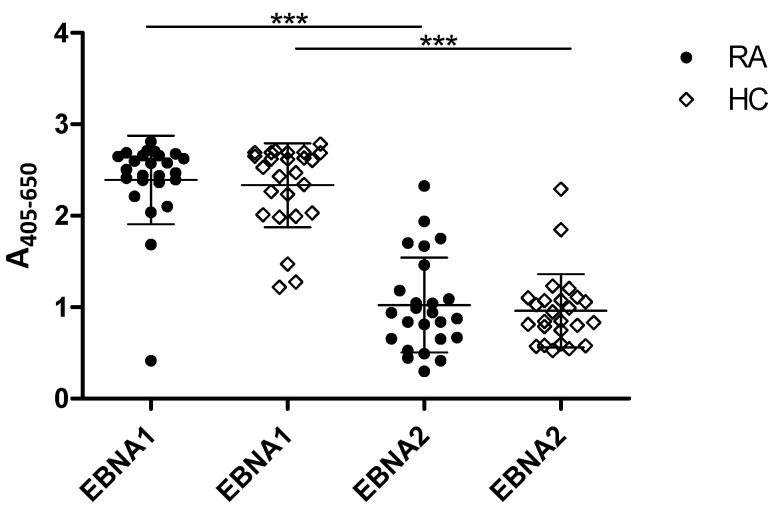
Reactivity of rheumatoid arthritis and healthy control samples to EBNA1 and EBNA2 full-length proteins analyzed using enzyme-linked immunosorbent assay. *** = *p* < 0.001.

**Figure 4 antibodies-11-00020-f004:**
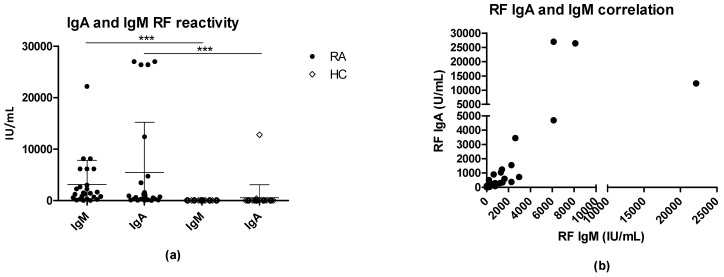
Determination of rheumatoid factor IgM and IgA in rheumatoid arthritis (RA) and healthy control (HC) samples by enzyme-linked immunosorbent assay: (**a**) RF IgA and IgM reactivity in RA and HC samples; (**b**) correlation between RF IgA and IgM. *** *p* < 0.001.

**Figure 5 antibodies-11-00020-f005:**
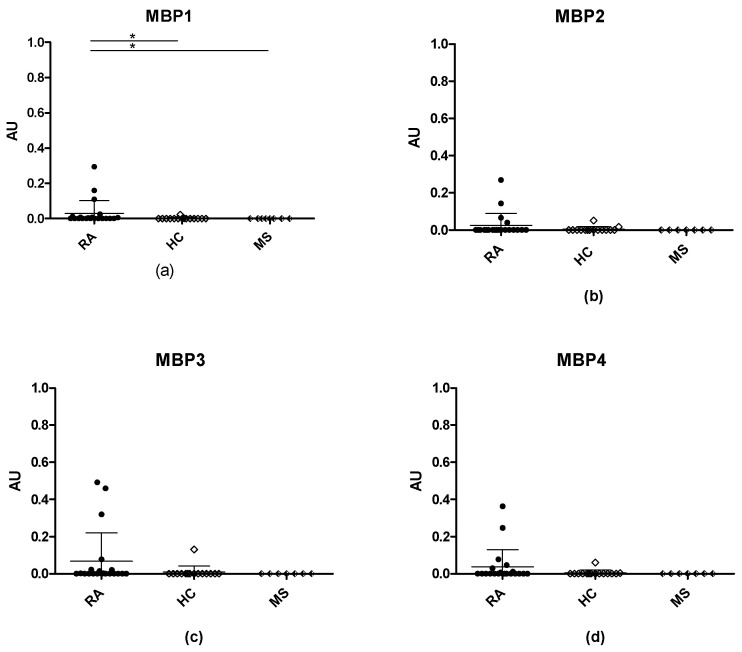
Reactivity of sera from rheumatoid arthritis (RA), multiple sclerosis (MS), and healthy controls (HC) to citrullinated MBP peptides analyzed by enzyme-linked immunosorbent assay: (**a**) reactivity of RA, MS and HC sera to MBP1; (**b**) reactivity of RA, MS and HC sera to MBP2; (**c**) reactivity of RA, MS and HC sera to MBP3; (**d**) reactivity of RA, MS and HC sera to MBP4. * *p* < 0.05.

**Table 1 antibodies-11-00020-t001:** Synthetic citrullinated peptides tested for antibody reactivity.

Origin	Name	Sequence
EBNA1	EBNA1-A	A-Cit-GGSRERARGRGRGRGEKR
	EBNA1-B	ARGGSRERARGRGRG-Cit-GEKR
	EBNA1-C	GGSKTSLYNLR-Cit-GTALAIPQ
EBNA2	EBNA2-A	GQGRGRWRG-Cit-GRSKGRGRMH
	EBNA2-B	GQSRGRGRG-Cit-GRGRGKGKSR
	EBNA2-C	KQGPDQGQG-Cit-GRWRGRGRSK
	NCit-EBNA2	G-Cit-GSKGRGRMHKLPEPRRPGPD
EBNA3	EBNA3-A	EPDSRDQQC-Cit-GQRRGDENRG
	EBNA3-B	PNENPYHAR-Cit-GIKEHVIQNA
	EBNA3-C	DQLPGVPKG-Cit-GACAPVPALA
	EBNA3-D	EDAHLEPSQ-Cit-GKKRKRVDDD
	EBNA3-E	AQAWNAGLL-Cit-GRAYGQDLL
	EBNA2-EBNA3	GQGRGRWRG-Cit-GQRRGDENRG
	EBNA2-EBNA2	EPDSRDQQS-Cit-GRSKGRGRMH
MBP	MBP1	KASTNSETN-Cit-GESEKKRNLG
	MBP2	SIGRFFGGD-Cit-GAPKRGSGKD
	MBP3	FGGDRGAPK-Cit-GSGKDSHHPA
	MBP4	TPPPSQGKG-Cit-GLSLSRFSWG

EBNA: Epstein-Barr nuclear antigen, MBP: Myelin basic protein.

**Table 2 antibodies-11-00020-t002:** Serologic reactivities to EBNA2-A, RF IgM and RF IgA.

	Single Positive				Double Positive			Triple Positive
Reactivities	None	EBNA2	RF IgM	RF IgA	RF IgM/A	EBNA2 + RF IgA	EBNA2 + RF IgM	RF IgA/M + EBNA2
Number (*n*)	0	19	24	23	23	18	19	17
Percentage (%)	0	76	96	88	88	72	76	68

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
