# Peer review of "Reactivity of Rheumatoid Arthritis-Associated Citrulline-Dependent Antibodies to Epstein-Barr Virus Nuclear Antigen1-3"

_2073-4468, 2022, doi:10.3390/antib11010020_

Round 1

Reviewer 1 Report

  1. How about the prevalence of positive anti-CCP in other autoimmune diseases except RA?
  2. In the part of discussion, the pathogenesis of anti-CCP production should be discussed.
  3. How about the activation of PAD? Mucosa? Bronchus ? Lung ?
  4. How about the association between EBV and autoimmune diseases except RA?
  5. SLE was associated with the EBV infection, and how about the presentations of these peptides in the patients with SLE
  6. In the part of discussion, the author should discuss the difference between different tools of diagnosis for RA including anti-CCP, EBNA2-A, RF.., etc.

Author Response

Dear reviewer,

Thank you for the constructive comments raised by you. All comments have been incorporated in the manuscript and have notably improved the draft. 

Please find our comments to the individual points below.

Best regards

Nicole Trier and Gunnar Houen 

Reviewer 1:

1. How about the prevalence of positive anti-CCP in other autoimmune diseases except RA?

Answer: The seropositive prevalence for ACPA in other autoimmune diseases have been elaborated in the introduction and in the discussion.

2. In the part of discussion, the pathogenesis of anti-CCP production should be discussed.

Answer: The pathogenesis of ACPA has been discussed in the end of the discussion.

3. How about the activation of PAD? Mucosa? Bronchus ? Lung ?

Answer: The activation of PAD by smoking and bacteria has been mentioned in the discussion.

4. How about the association between EBV and autoimmune diseases except RA?

Answer: The association between EBV and other autoimmune diseases has been elaborated in the introduction.

5. SLE was associated with the EBV infection, and how about the presentations of these peptides in the patients with SLE.

Answer: The EBNA1 peptides and EBNA2-A has already been tested for reactivity to sera from Sjöogren’s syndrome and systemic lupus erythematosus in another study. This has been elaborated in the discussion section.  

6. In the part of discussion, the author should discuss the difference between different tools of diagnosis for RA including anti-CCP, EBNA2-A, RF.., etc.

Answer: Amended as requested. RF and CCP assays have been elaborated in the discussion section.  

Reviewer 2 Report

The topic of rheumatoid arthritis diagnostics is an extremely serious problem, because the optimal method of diagnosing this disease has not been developed to date. High hopes are attached to ACPA as a disease marker. Therefore, I believe that the research conducted in this article is important. However, I have some suggestions and ambiguities that I would like to verify.

Please explain why this idea for the interpretation of the ELISA test came from, wouldn't it be better to present the results that are not normalized to the pooled serum? Especially since there can be both sero positive and seronegative patients in the RA patient group?

As far as editorial issues are concerned, I would suggest to mark HD with a different sign, eg a triangle, because in some figures it is a problem, unlike RA from HD, eg Fig 4.

Author Response

Dear reviewer.

Thank you for the constructive comments. All comments have been incorporated in the manuscript and have notably improved the draft. 

Please find our comments to the individual points below.

Best regards

Nicole Trier and Gunnar Houen 

Reviewer 2:

The topic of rheumatoid arthritis diagnostics is an extremely serious problem, because the optimal method of diagnosing this disease has not been developed to date. High hopes are attached to ACPA as a disease marker. Therefore, I believe that the research conducted in this article is important. However, I have some suggestions and ambiguities that I would like to verify.

Please explain why this idea for the interpretation of the ELISA test came from, wouldn't it be better to present the results that are not normalized to the pooled serum? Especially since there can be both sero positive and seronegative patients in the RA patient group?

Answer: The pool used for normalization only contained ACPA-positive samples, this has been described in the materials and methods section. Results were normalized in order to be able to compare results directly, as assay variations occasionally may interfere.

As far as editorial issues are concerned, I would suggest to mark HD with a different sign, eg a triangle, because in some figures it is a problem, unlike RA from HD, eg Fig 4.

Answer: The figures have been adjusted and new symbols for HDs have been selected.

Round 2

Reviewer 1 Report

Well response from the author